# Intraoperative cerebral oximetry in open heart surgeries reduced postoperative complications: A retrospective study

**Norsham Juliana**[1]*, **Noor Anisah Abu Yazit**[1], **Suhaini Kadiman**[2], **Kamilah Muhammad Hafidz**[2], **Sahar Azmani**[1], **Nur Islami Mohd Fahmi Teng**[3‡], **Srijit Das**[4‡]

**1** Faculty Medicine and Health Science, Universiti Sains Islam Malaysia, Nilai, Negeri Sembilan, Malaysia, **2** Anaesthesiology Department and Intensive Care, National Heart Institute, Kuala Lumpur, Malaysia, **3** Faculty of Health Science, Universiti Teknologi MARA, Puncak Alam, Malaysia, **4** Department of Human & Clinical Anatomy, College of Medicine & Health Sciences, Sultan Qaboos University, Muscat, Oman

☯ These authors contributed equally to this work.
‡ These authors also contributed equally to this work.
* njuliana@usim.edu.my

**Data Availability Statement:** All relevant data are within the paper.

## Abstract

Cardiothoracic surgeries are life-saving procedures but often it results in various complications. Intraoperative cerebral oximetry monitoring used to detect regional cerebral oxygen saturation (rScO2) is a non-invasive method that provides prognostic importance in cardiac surgery. The main aim of the present study was to evaluate the association of intraoperative cerebral oxygen monitoring during cardiac surgery on postoperative complications. This was a case-controlled retrospective study conducted on adult patients, who underwent open-heart surgery in National Heart Institute, Malaysia. The case group comprised patients on protocolized cerebral oximetry monitoring. They were treated using a standardized algorithm to maintain rScO2 not lower than 20% of baseline rScO2. The control group comprised patients with matched demographic background, preoperative risk factors, and type of surgical procedures. Propensity score stratification was utilized to contend with selection bias. Retrospective analysis was performed on 240 patients (case group) while comparing it to 407 patients (control group). The non-availability of cerebral oximetry monitoring during surgery was the prominent predictor for all outcome of complications; stroke (OR: 7.66), renal failure needing dialysis (OR: 5.12) and mortality (OR: 20.51). Postoperative complications revealed that there were significant differences for risk of mortality ($p < 0.001$, OR = 20.51), renal failure that required dialysis ($p < 0.001$, OR = 5.12) and stroke ($p < 0.05$, OR = 7.66). Protocolized cerebral oximetry monitoring during cardiothoracic surgeries was found to be associated with lower incidence of stroke, renal failure requiring dialysis and mortality rate.

## Introduction

Cardiothoracic surgeries are life-saving procedures but its complexity results in various complications. In general, cardiothoracic patients comprise the elderly and those who have

**Funding:** The author(s) received no specific funding for this work.

**Competing interests:** The authors have declared that no competing interests exist.

multiple concomitant comorbidities. Therefore, they are susceptible to various adverse effects of the surgery [1]. Despite the advancement of modern technology, the prevalence of brain injury in post-surgery remains significantly high. One of the serious detrimental complications is the stroke that occurs in 10% among the susceptible subset of patients [2]. Appropriate and timely management of postoperative complications is imperative in reducing the risk of morbidity and mortality among them. Despite the modern intervention, postoperative stroke remains an independent risk factor for postoperative mortality [3].

Conclusive theories behind the incidence of cerebral injury following cardiothoracic surgeries are still debatable. The underlying mechanism possibly includes an occurrence of embolization, hypoperfusion that leads to cerebral ischemia or brain hyperperfusion that leads to cerebral edema. Hence, continuous monitoring of cerebral oxygen saturation may provide the solution to reduce the incidence of hypoperfusion or hyperperfusion and eventually reduce undesirable complication of the surgery. Previous studies proposed that continuous monitoring of cerebral oxygenation allows early detection of cerebral hypoxia, thereby permitting early intervention to restore parameters during cardiac surgery [4, 5]. Furthermore, continuous monitoring improves clinical decisions, shortens the length in intensive care unit and hospital stay. Despite an additional device being needed in the surgical theatre that may increase the surgical cost, all possible benefits are a part of decreasing financial expenses, while dealing with post-surgical complications [6]. Another important concern highlighted in earlier studies was the difficulty to establish the specificity of $rScO_2$ monitoring in ensuring cerebral perfusion. Previous studies also reported poor relationship between cerebral desaturations and postoperative neurological complication (PNC), however, the result was explained by the fact that low $rScO_2$ occurred only sporadically in their patient population [7].

Near-infrared spectroscopy (NIRS) based cerebral oximetry is a non-invasive technique that generates regional cerebral oxygen saturation ($rScO_2$) [8]. Its measurement is based on different absorption capacities of oxygenated, and deoxygenated hemoglobin with oxygenated hemoglobin ($HbO_2$) that absorbs more infrared light than deoxygenated hemoglobin. The technique suits the continuous monitoring as it is non-invasive and reflects the balance between cerebral oxygen supply and demand [9]. The present study aimed to evaluate the association of intraoperative cerebral oxygen monitoring during cardiac surgery on the complications.

## Materials and methods

### Study design

This was a case-controlled study conducted on adult patients who underwent open-heart surgery in National Heart Institute (NHI), Malaysia. The study was approved by the National Heart Institute Research Ethics Committee. The patients were selected from Cardiothoracic Surgery Registry and IntelliVue Clinical Information Portfolio. The data was retrieved on 9[th] December 2019 and was fully anonymized. All enrolled subjects were high-risk patients and those who underwent open heart surgeries from the year 2017 until 2019. Written informed consent was obtained from all subjects prior to the surgery. Cases were among those on cerebral oximetry monitoring intraoperatively, and the control included those without monitoring. The inclusion criteria were all adult patients who underwent open heart surgeries from 2017 to 2019 which utilized cerebral oximetry based on the high-risk profile and clinical judgement. Patients with pre-existing comorbidities of diabetes mellitus, renal impairment, and previous stroke were categorized as high-risk profile patients. The presence of all these conditions represented additional risk factors for major surgical procedures. Above all, the procedures involving coronary artery bypass graft, valvular heart (mitral, aortic, double valve, etc.) and

other heart surgeries itself carry its own independent risks. Hence, the intertwine of patients' and procedures risk factors intensified the risks for postoperative complications for the study population.

## Patient management

Cerebral oximetry sensors were applied on the forehead before the induction of anesthesia and patient's baseline, and continuous cerebral oxygenation was recorded throughout the surgery. The monitoring was discontinued in the intensive care unit (ICU) after the patient was separated from the ventilator. The monitoring device that was used in this study was In Vivo Optical Spectroscopy (INVOS)® (Somanetics/Covidien, Inc., Boulder, CO, USA). The system uses NIRS technique for non-invasive and continuous measurement of oxygen saturation. All patients on INVOS were treated using the algorithm to maintain $rScO_2$ baseline value recorded before the induction of anesthesia at room air. If the $rScO_2$ value during surgery decreased below 20% of baseline $rScO_2$, standardized interventions were taken immediately by the clinicians to maintain the $rScO_2$ above those values. The interventions include repositioning of head or bypass cannula to eliminate mechanical obstruction, increasing cerebral oxygen delivery, or reducing cerebral oxygen consumptions. For those patients under monitoring with persistent $rScO_2$ below treatment threshold, the $FiO_2$ was increased to 100%, the $PaCO_2$ was optimized at 45mmHg, and the hematocrit was increased to at least 25%. Apart from the standardized algorithm in those patients on $rScO_2$ monitoring, the rest of the surgical and the anesthesia management of the patients followed the institution's standard of care. Fig 1 describes the patient's management based on the cerebral oximetry.

## Data analysis

Important parameters for postoperative complications such as in-hospital stroke, delirium, renal failure required dialysis, arrhythmias and mortality within the two groups were observed and compared. Apart from that, the parameters for patients' duration of hospital stay were also compared.

Continuous data was presented as mean ± standard deviation, while categorical data was presented as frequency (%). The data was analyzed using Statistical Package for Social Science (SPSS) software version 23 (SPSS Inc., Chicage, IL). Independent t-test was used to compare all the continuous quantitative data. In order to identify putative predictor variables for logistic regression, variables were first screened on univariate analysis based on either occurrence of medical or surgical complications. Variables were entered with a stepwise backward likelihood method. Odds ratios (ORs) were shown with 95% confidence intervals (CIs). The level of significance was set at $p < 0.05$.

## Results

A total of 240 patients who fulfilled the inclusion and exclusion criteria were selected as cases (Case group). Comparative data was collected from 407 (63%) patients, as controls with propensity scoring matched to the case group. The mean age of the patients was 56.7 ± 13.5 years. In the case group, there were 168 males and 72 females, while the control group included 289 males and 118 females. Both the case and control groups showed homogeneity in their demographic data, Euroscore, duration of cardiopulmonary bypass time, preoperative risk factors, and type of procedures, with no significant difference being observed ($p > 0.05$). The comparisons between the case and control groups for demographic profiles, other risk factors and postoperative complications were shown in Table 1.

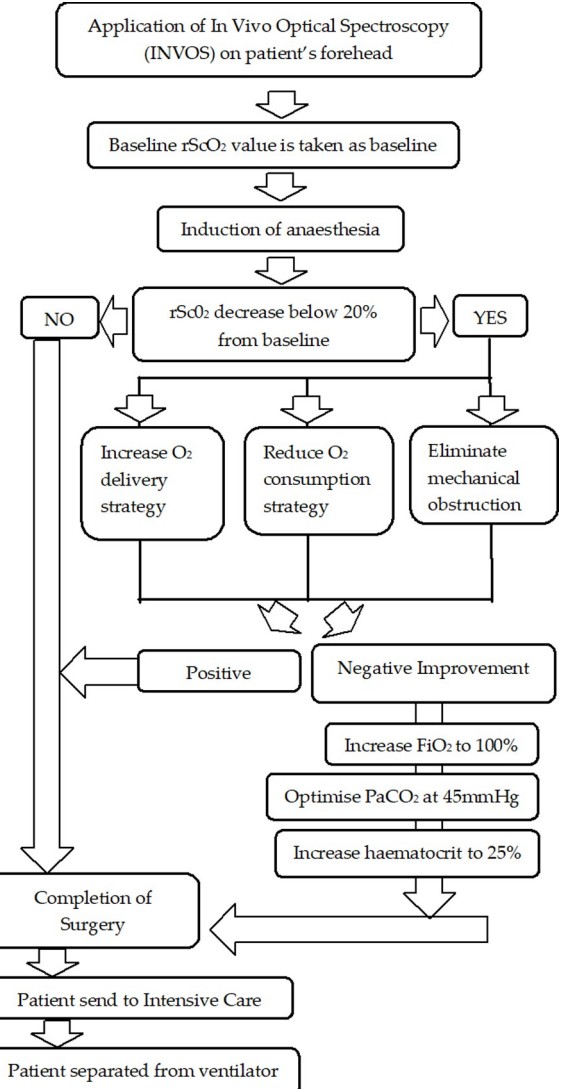

**Fig 1. Patient management based on cerebral oximetry.**

Overall, rate of postoperative complications, specifically for in-hospital stroke (p<0.05), renal failure need dialysis (p<0.001) and mortality (p<0.001) were significantly higher in the control group compared to the case group.

Logistic regression was conducted to further analyze the predictors involved that contributed to each significant complication. The non-availability of cerebral oximetry monitoring during surgery was the prominent predictor for all outcome of complications; stroke (OR: 7.66), renal failure needing dialysis (OR: 5.12) and mortality (OR: 20.51). Table 2 summarizes the predictors for all three significant complications that differ between the case and control groups. Univariate analysis was performed independently for each selected predictor. Those in the control group (patients without cerebral oximetry monitoring) proved to have increased odds of having all the main postoperative complications as compared to those in the case group (patients on cerebral oximetry monitoring via INVOS). Hence, multivariate analysis was performed by including all significant predictors.

**Table 1. Demographic profiles, risk factors and postoperative complications of the case and control groups.**

| Variables | Group | | *p*-value |
|---|---|---|---|
| | Case (n = 240) | Control (n = 407) | |
| Age (years) | 57.0 ± 12.8 | 56.5 ± 13.8 | 0.752 |
| Age categorical (% patients ≥60 years) | 50.8 | 55.8 | 0.223 |
| Gender (%male/female) | 70/30 | 69/31 | 0.798 |
| EuroSCORE (%) | 5.25 ± 2.97 | 4.91 ± 3.35 | 0.053 |
| CPBypass time (% >2 hours) | 41.9 | 40.7 | 0.779 |
| Preoperative Risk Factors % (n) | | | |
| Hypertension | 74.6 (179) | 70.1 (285) | 0.233 |
| Diabetes | 40.1 (96) | 40.2 (163) | 0.987 |
| Hypercholerolemia | 65.5 (157) | 59.7 (243) | 0.154 |
| Smoking | 42.2 (101) | 42.4 (173) | 0.974 |
| Synus rhythm | 81.7 (196) | 81.1 (330) | 0.854 |
| Chronic kidney disease | 10.0 (24) | 15.2 (62) | 0.058 |
| Chronic renal failure | 4.6 (11) | 7.1 (29) | 0.195 |
| Cerebrovascular accident | 10.0 (24) | 9.1 (37) | 0.702 |
| Procedure (%) | | | |
| All CABG | 60.0 (144) | 59.0 (240) | 0.209 |
| All valve | 26.7 (64) | 31.4 (128) | |
| Others | 13.3 (32) | 9.6(39) | |
| Bypass time (hours) | 2.0 ± 0.9 | 1.95 ± 0.96 | 0.481 |
| Duration of ventilation (hours) | 34.0 ± 69.90 | 28.26 ± 32.4 | 0.820 |
| Duration of ICU stay (days) | 3.3 ± 4.6 | 3.55 ± 5.43 | 0.137 |
| Duration of post-op stay (days) | 11.79 ± 14.1 | 11.42 ± 14.15 | 0.087 |
| Postoperative complications (%) | | | |
| Stroke | 1.4 (3) | 5.1 (21) | 0.026* |
| Renal failure need dialysis | 8.1 (19) | 24.1 (98) | <0.001* |
| Arrythmia | 31.0 (74) | 36.0 (147) | 0.223 |
| Arrythmia, AF | 24.3 (58) | 24.1 (98) | 0.885 |
| Mortality | 7.9 (19) | 28.7 (117) | <0.001* |

Data are expressed as frequencies % or mean ± standard deviation. CABG: coronary artery bypass graft, ICU: intensive care unit.

*data is significant at *p*<0.05.

## Discussion

Globally, cardiac surgeries emerged as an essential component of global health. Despite current advancement, cardiac surgery remains a very complex area for outcome prediction. Based on the Society for Thoracic Surgeons (STS), undesirable in-hospital mortality after cardiac surgery accounts for 1–4% of all types of cardiac surgery [10]. The patients selected for this retrospective study were among high-risk patients with pre-existing multiple co-morbidities, having a higher Euroscore, and some of them underwent a high-risk surgery. Examples of high-risk surgery were double valve surgery and thoracic aorta surgery, with 40% of them had the duration of cardiopulmonary bypass time of more than 2 hours. Therefore, the higher mortality rate observed among patients in this study was explained. This finding was in accordance with other studies on cardiac surgeries that utilized high-risk patients as the sampling frame [11, 12].

In order to assure the homogeneity between the case and control group in this study, their demographic data, preoperative risk factors, Euroscore, and type of cardiac surgery were carefully matched [13, 14]. In addition, the duration of in-hospital stay together with the duration of

**Table 2. Predictors for significant complications in case and control group.**

| Variables | Univariate logistic regression OR (95% CI) | *p*-value | Multivariate logistic regression OR (95% CI) | *p*-value |
|---|---|---|---|---|
| **Postoperative Stroke** | | | | |
| Gender (male) | 4.48 (1.04–19.39) | 0.045 | | |
| Hypercholesterolemia | 3.95 (1.16–13.52) | 0.029 | | |
| Bypass time >2 hours | 2.67 (1.08–6.53) | 0.033 | | |
| ICU stay | 1.09 (1.04–1.15) | 0.001 | | |
| Control (Not using INVOS) | 3.70 (1.08–12.64) | 0.037 | 7.66 (0.97–60.42) | 0.053 |
| **Renal Failure Need Dialysis** | | | | |
| Age | 1.94 (1.26–2.97) | 0.003 | | |
| Hypertension | 3.89 (2.02–7.49) | <0.001 | | |
| Diabetes | 3.88 (2.47–6.08) | <0.001 | | |
| Hypercholesterolemia | 4.51 (2.50–8.17) | <0.001 | | |
| CKD | 8.23 (4.92–13.77) | <0.001 | | |
| CRF (dialysis) | 13.46 (5.99–30.23) | <0.001 | | |
| ICU stay | 1.20 (1.11–1.29) | <0.001 | | |
| Control (Not using INVOS) | 3.63 (2.10–6.29) | <0.001 | 5.12 (1.68–15.55) | 0.004* |
| **Mortality** | | | | |
| Age >60 years | 2.24 (1.52–3.30) | <0.001 | | |
| Hypertension | 2.35 (1.42–3.88) | <0.001 | | |
| Diabetes | 2.38 (1.60–3.52) | <0.001 | | |
| Hypercholesterolemia | 2.27 (1.46–3.53) | <0.001 | | |
| CKD | 4.74 (2.94–7.64) | <0.001 | | |
| Bypass time >2 hours | 2.84 (1.92–4.19) | <0.001 | | |
| Duration ventilation | 1.0 (1.00–1.01) | 0.017 | | |
| Control (Not using INVOS) | 4.69 (2.80–7.86) | <0.001 | 20.51 (3.32–126.64) | <0.001* |

Univariate logistic regression analysis was performed for all listed variables and only statistically significant results are presented in the table.

Multivariate logistic regression analysis: *indicates data is significant at *p*<0.05.

OR: odds ratio; 95% Cl: 95% confidence interval; ICU: intensive care unit; INVOS: IN Vivo Optical Spectroscopy; CKD: chronic kidney disease; CRF: chronic renal failure.

ICU stay was also analyzed and matched between both groups (p>0.05). The propensity scoring and matching are important as Mazeffi et al. [3] pointed out that most in-hospital morbidity and mortality are influenced by the underlying patients and surgical factors and they tend to occur during the first week following cardiac surgery. Therefore, the increased length of hospital and ICU stay are also linked to increased likelihood of in-hospital morbidity and mortality.

The case group in this study that received cerebral oximetry monitoring, exhibited a reduction in the rate of post-surgical complications. Utilization of cerebral oximetry was proposed as one of the potential methods to reduce the adverse events. However, clear evidence is still lacking with previous studies showing conflicting results [14–17]. Existing studies debated on the effect of cerebral oximetry monitoring on neurocognitive function following cardiac surgery. Salter et al. [15] found that there was no significant association between the use of a cerebral oximetry guided intervention in reducing the incidence of postoperative cognitive dysfunction. On the other hand, Colak et al. [14] found that postoperative cognitive outcome was significantly better in patients with intraoperative cerebral oximetry monitoring. Recent studies also showed that cerebral oximetry monitoring was capable in reducing the postoperative delirium following cardiac surgery [18, 19]. The conflicting results between existing studies, warrant serious attention for further intervention in order to provide solid evidence on the

benefits and importance of cerebral oximetry monitoring. Nevertheless, maintaining cerebral perfusion is critical, preoperatively. Interventions were only possible in the group utilised the $rScO_2$ as a monitor of adequacy of cerebral perfusion. In the group that does not utilised $rScO_2$, the interventions were unlikely to be executed as there were no parameters to trigger nor guide the effectiveness of interventions. The patients who managed to escape complications of stroke and its associated mortality risk, as one far extreme of the spectrum of brain injury post-surgery, may suffer another undesirable complication of postoperative cognitive decline related to cerebral hypoperfusion, as another extreme of the spectrum of the brain injury post-surgery. Multiple trials were tailor-made to explain the mechanism of neurocognitive injury that occurred among post cardiac surgery patients. Unravelling the role of procedural (anaesthesia, surgical technique, length of surgery, etc.) and patient (age related cognitive decline, underlying comorbidities, etc.) factors always remained debatable [20]. Findings from all these studies showed that there was a need for in-depth pathophysiological explanation by elucidating genetic and proteomic approaches [20, 21].

The odds of having the postoperative stroke (OR = 3.70; AOR = 7.66) was profound in the control group compared to the case group. Ischemic stroke related to perioperative cerebral hypoperfusion injury is a well-recognized complication in cardiac surgery. Recent advancement in surgical techniques has progressively reduced stroke rates to 1–4%. However, patients with diabetes have a higher risk of post-cardiac surgery stroke and the risk is also increased linearly with advancement in age [21]. Retrospective studies reported the relationship between $rScO_2$ desaturation and stroke or type I and II neurologic injury following surgeries. However, the studies had multiple limitations including small sample size (59 subjects and 46 subjects) and the assessments were only made during the immediate postoperative period [22]. A large retrospective study by Goldman et al. [23] monitored 1034 patients cerebral regional cerebral $O_2$ saturation ($rScO_2$) and found that the stroke rate was lower in the monitored patients compared to 1245 historical controls without monitoring.

In symphony with the finding of postoperative stroke, renal failure that required dialysis (OR = 3.63; AOR = 5.12) was also found to be higher in the control group. Kidney injury following cardiac surgery that occurred in 5% to 30% of patients was hypothesized to result from kidney hypoperfusion. Adverse events on the kidney predisposes the post cardiac surgery patients to a risk of in-hospital and long-term mortality. Monitoring of cerebral oxygenation was found to provide a novel method for precisely guiding mean arterial blood pressure (MAP) targets during cardiac surgery. Results from this study was in agreement with previous studies that showed cerebral oximetry monitoring provides better outcome on the kidney following cardiac surgery probably by improving the regional perfusion to the kidney during cardiopulmonary bypass [24–26].

Mortality rate was found to be lower in the case group with only 7.9% compared to 28.7% in the control group. The result was in accordance with another study that highlighted the importance of cerebral oximetry to monitor brain as an index organ, thereby reducing the outcome of mortality after cardiac surgery [27]. Intraoperative continuous information about brain oxygenation allows the use of brain as sentinel organ indexing overall organ perfusion and injury, thus, reducing the risk of mortality [6]. Four randomized controlled trials including a systematic review did not show any significant difference between the intervention group and the control group [22]. Hence, we need more interventional studies to provide evidence in assisting clinical judgment to recommend cerebral oximetry monitoring as a standard protocol. The major highlight of this study was the novel use of $rScO_2$, in addition to other standard monitoring parameters used in cardiac surgery. This is important to guide the clinical decision to intervene either by or including alleviating possible blood flow obstruction from repositioning the head or aortic cannula, improving $O_2$ delivery, and reducing the $O_2$ consumption, using the brain as the indexed organ.

However, the strength of this study lies on the fact that the homogeneity characteristics of patients' selection that only focused on high-risk patients. Moreover, propensity scores were calculated to reduce selection bias between both groups [28]. Data in this study were limited to retrospective data, hence the extent of underlying co-morbidities were limited to available data in patients' medical record [29]. There were some key parameters which could not be measured together in this study i.e. patients' details in depth of each pre-operative parameter in order to stratify the severity of their pre-existing conditions. Moreover, data from this study included all types of surgical techniques in heart surgery and variety of heart surgeons. As this study pointed out the benefits of cerebral oximetry among high-risk patients going for high-risk surgeries, future studies may focus on analyzing its benefit based on specific surgical technique. Therefore, we may be able to understand and determine the type of surgery that should be prioritized to benefit from the cerebral oximetry monitoring.

Continuous monitoring of $rScO_2$ in the ICU post-surgery is only implemented in critical situation for patients with high risk towards detrimental neurological outcome such as those with suspected brain injury that requires cerebral resuscitation. Nevertheless, the most likely point of injuries to the brain occurs intraoperatively rather than in the post-operative period [30].

## Conclusion

Despite the clear evidence of equipoise with respect to the benefits of cerebral oximetry monitoring clinically, the monitoring is still an optional method at cardiac surgery centers, worldwide. The results support the hypothesis that the usage of cerebral oximetry monitoring gave a significant impact in reducing after cardiac surgery complications among high-risk patients going for high-risk surgeries. In order to determine the clinical utility of the method, future randomized control trials should be conducted in order to evaluate clinically important outcomes that had been described in this retrospective study.

## Acknowledgments

The authors acknowledge the help received from Norfazlina Jaffar from Clinical Research Department, National Heart Institute for data extraction from software.

## Author Contributions

**Conceptualization:** Norsham Juliana, Suhaini Kadiman.

**Data curation:** Kamilah Muhammad Hafidz.

**Formal analysis:** Noor Anisah Abu Yazit.

**Investigation:** Suhaini Kadiman.

**Methodology:** Suhaini Kadiman, Kamilah Muhammad Hafidz.

**Project administration:** Suhaini Kadiman.

**Resources:** Sahar Azmani.

**Supervision:** Sahar Azmani, Srijit Das.

**Visualization:** Kamilah Muhammad Hafidz.

**Writing – original draft:** Norsham Juliana, Noor Anisah Abu Yazit, Sahar Azmani.

**Writing – review & editing:** Noor Anisah Abu Yazit, Nur Islami Mohd Fahmi Teng, Srijit Das.

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
