## [Decision Letter · Decision Letter 0]

21 Apr 2021

Intraoperative cerebral oximetry in open heart surgeries reduced postoperative complications: a retrospective study

PONE-D-20-35863

Dear Dr. Juliana,

We’re pleased to inform you that your manuscript has been judged scientifically suitable for publication and will be formally accepted for publication once it meets all outstanding technical requirements.

Kind regards,

Chun Chieh Yeh, M.D., Ph.D.

Academic Editor

PLOS ONE

Journal Requirements:

1.Please ensure that you include a title page within your main document. We do appreciate that you have a title page document uploaded as a separate file, however, as per our author guidelines (http://journals.plos.org/plosone/s/submission-guidelines#loc-title-page) we do require this to be part of the manuscript file itself and not uploaded separately.

2. In your ethics statement in the Methods section and in the online submission form, please provide additional information about the data used in your retrospective study. Specifically, please ensure that you have discussed whether all data from the  Cardiothoracic Surgery Registry and the IntelliVue Clinical Information Portfolio were fully anonymized before you accessed them.

3. Please include the date(s) on which you accessed the databases or records to obtain the data used in your study.

4. Please report your p-values in table 2.

Additional Editor Comments:

Thanks for your great work in this field that provide insightful information and message in this field. We are glad to accept this article at its current content.

Reviewers' comments:

Reviewer's Responses to Questions

**Comments to the Author**

1. Is the manuscript technically sound, and do the data support the conclusions?

Reviewer #1: Yes

Reviewer #2: Yes

2. Has the statistical analysis been performed appropriately and rigorously? 

Reviewer #1: Yes

Reviewer #2: Yes

3. Have the authors made all data underlying the findings in their manuscript fully available?

Reviewer #1: Yes

Reviewer #2: Yes

4. Is the manuscript presented in an intelligible fashion and written in standard English?

Reviewer #1: Yes

Reviewer #2: Yes

5. Review Comments to the Author

Reviewer #1: Dear the authors of the manuscript entitled "Intraoperative cerebral oximetry in open heart surgeries reduced postoperative complications: a retrospective study"

I was glad reading your manuscript which highlighted the high importance of implementing the introperative cerebral oximetry in decreasing the rate of major complications in cardiac surgery

I think the manuscript is well written and easy to understand by the reader, also there is good literature review and data analysis

I have no concerns

Reviewer #2: An excellent manuscript in its field, I have no further comments. In its current form, I suggest that it is accepted.

An excellent manuscript in its field, I have no further comments. In its current form, I suggest that it is accepted.

6. PLOS authors have the option to publish the peer review history of their article (what does this mean?). If published, this will include your full peer review and any attached files.

Reviewer #1: **Yes: **salah eldien Altarabsheh

Reviewer #2: No

---

## [Editor Report · Acceptance letter]

26 Apr 2021

PONE-D-20-35863 

Intraoperative cerebral oximetry in open heart surgeries reduced postoperative complications: a retrospective study 

Dear Dr. Juliana:

I'm pleased to inform you that your manuscript has been deemed suitable for publication in PLOS ONE. Congratulations! Your manuscript is now with our production department. 

Kind regards, 

on behalf of

Dr. Chun Chieh Yeh 

Academic Editor

PLOS ONE